# A presaccadic perceptual impairment at the postsaccadic location of the blindspot

**Daniel T. Smith**[ORCID]**\*, Ulrik Beierholm, Mark Avery**

Department of Psychology, Durham University, Durham, United Kingdom

\* daniel.smith2@durham.ac.uk

## Abstract

Saccadic eye movements are preceded by profound changes in visual perception. These changes have been linked to the phenomenon of 'forward remapping', in which cells begin to respond to stimuli that appear in their post-saccadic receptive field before the eye has moved. Few studies have examined the perceptual consequences of remapping of areas of impaired sensory acuity, such as the blindspot. Understanding the perceptual consequences of remapping of scotomas may produce important insights into why some neurovisual deficits, such as hemianopia are so intractable for rehabilitation. The current study took advantage of a naturally occurring scotoma in healthy participants (the blindspot) to examine pre-saccadic perception at the upcoming location of the blindspot. Participants viewed stimuli monocularly and were required to make stimulus-driven vertical eye-movements. At a variable latency between the onset of saccade target (ST) and saccade execution a discrimination target (DT) was presented at one of 4 possible locations; within the blindspot, contralateral to the blindspot, in post-saccadic location of the blindspot and contralateral to the post-saccadic location of the blindspot. There was a significant perceptual impairment at the post-saccadic location of the blindspot relative to the contralateral post-saccadic location of the blindspot and the post-saccadic location of the blindspot in a no-saccade control condition. These data are consistent with the idea that the visual system includes a representation of the blindspot which is remapped prior to saccade onset.

**Data Availability Statement:** https://osf.io/c47te/.

**Funding:** The author(s) received no specific funding for this work.

## Introduction

Effective visual exploration requires saccadic eye movements that rapidly move the foveal region to sample different parts of the environment. An enduring question concerns the mechanisms that allow effective control integration of visual information across these eye-movements [1, 2]. One influential proposal is that the visuo-motor system makes predictions about the sensorimotor consequences of an action [3] and contrasts predicted sensory input with actual inputs. If the actual sensory input following an action matches these predictions, the system concludes the action has been correctly executed and the participant experiences seamless visual continuity.

The idea that the visual system engages in predictive coding is consistent with evidence of anticipatory responses in the visual system. For example, in the moments before a saccadic

**Competing interests:** The authors have declared that no competing interests exist.

eye-movement the focus of attention is shifted to the upcoming location of fovea [4, 5] and if the scene contains important but non-foveal objects presaccadic cues can facilitate processing at the post-saccadic retinal location of the object, even before saccade onset [6–8]. Anticipatory responses can also be observed at a neuronal level. In the phenomenon of forward remapping, some cells responding to objects that appear at the post-saccadic location of their receptive field (the future field; FF) before the saccade has been released [9]. These changes begin around 250ms before the onset of the saccade that will bring the object into their classical receptive field [10]. Predictive remapping effects are evident at multiple levels of the primate visual system, including the Superior Colliculus [11], extrastriate cortex [12, 13], Lateral Intraparietal Area [9] and Frontal Eye Fields [14], although the number of cells showing the effects decreases at progressively lower levels of the visual system, with ~35% LIP neurons showing presaccadic remapping compared to only 16% of V3A neurons and ~2% of V1 neurons [12, 15].

These studies provide strong evidence that the visual system makes use of predictions to support perceptual stability, but the precise relationship between behavioural remapping effects and neurophysiological remapping effects remains contentious. Indeed, it has been argued that there may be several related but separable types of remapping [2], including the classical 'forward' remapping [9], in which receptive fields remap along the vector of the saccade, 'convergent' remapping, in which cells remap towards the saccade goal (e.g. [16–18]), and attentional remapping [19] which explains remapping in terms of anticipatory attentional activation of cells that will receive visual input after the saccade is completed.

Studies of remapping typically examine anticipatory responses when the saccade will bring the object of interest into a sighted part of the visual field. This makes sense if one assumes that the function of remapping is to maintain perceptual stability. A largely neglected issue is to what extent the visual system can make predictions about a predictable loss of visual information, as occurs when a part of the visual field lands on the retinal blindspot after a saccade. This is a relatively trivial issue in the healthy visual system, as the area of blindness around the optic nerve is small, peripheral and compensated for by the contralateral eye under binocular viewing conditions. However, during monocular vision, and in patients with neuro- visual field defects such as hemianopia the potential for the system to predict the imminent loss of visual information due to a saccade is more important. In the latter case a better understanding of how the visual system processes scotoma may help explain why these deficits are so resistant to compensatory or restorative therapy [20, 21].

The current study therefore examined perceptual performance at the current and future location of blindspot. The blindspot is represented in visual cortex by relatively large receptive fields that do not receive input from sensory cortex [22]. If these receptive fields are subject to presaccadic forward remapping the blindspot may shift to its post-saccadic location in the moments before the saccade is executed (its 'future field'). In this case, the ability to discriminate the orientation of a target presented at the future location of the blindspot should be impaired. This prediction was tested by comparing presaccadic discrimination accuracy at the blindspot future field (FF) with discrimination accuracy at 3 control locations; the blindspot (Blindspot), contralateral to the blindspot (Blindspot Contra) and contralateral to the blindspot future field (FF Contra). Performance was also tested in Fixate condition in which no saccade was executed.

## Materials and methods

### Participants

Eight participants (4 female, 8 right-eyed, ages 19–38) were recruited from Durham University during the 2016/2017 academic year. All participants gave informed written consent. Authors

had access to information that could identify individual participants during collection. The experiment was approved by the Durham University Department of Psychology Research Ethics Committee (Ref 15/44) and was conducted in accordance with the BPS guidelines for experimental research in psychology.

## Stimuli and apparatus

The discrimination target was a white bar (0.3˚ x 0.9˚) presented on a black background. The mask was a black and white checkerboard (1˚ x 1˚). The centre of discrimination target was at one of four locations relative to fixation: 15˚ temporal, 1˚ inferior (the blindspot); 15˚ nasal, 1˚ inferior; 15˚ temporal, 4.8˚ superior; 15˚ nasal, 4.8˚ superior. The fixation point was a white spot (0.1˚) located 5.8˚ inferior to the horizontal midline of the monitor. The saccade target was a white box (0.1˚ x 0.1˚), presented 5.8˚ superior to the fixation point on the vertical meridian. Stimuli were generated using a Cambridge Research Systems ViSaGe graphics card and displayed on a 17-inch Sony Trinitron CRT monitor with a refresh rate of 100 Hz. Responses were collected using a two-button response box. Fixation was monitored using a Cambridge Research Systems Videoeyetracker Toolbox sampling at 250 Hz. The experiment was conducted in the dark.

## Perimetric blindspot assessment

The location of the blindspot was established with an Oculus Twinfield 2 automatic perimeter (Oculus Optikgerate GmbH, Wetzlar-Dutenhofen, Germany), with a background luminance of 10 cd/m$^2$. The mapped area extended from 5˚ to 19˚ into the temporal hemispace and 5˚ superior to 5˚ inferior of the horizontal midline (see Fig 1 for a representative example a field plot). Participants placed the head in a chinrest and were instructed to fixate the central red fixation points. They pressed a buzzer when they detected a stimulus. The probe was a static, supra threshold white spot (Goldman III) displayed for 200ms. Probe locations were separated by 2˚. On average the blindspot subtended 5˚ vertically and 3.5˚ horizontally. The mean distance of the probe from the upper and lower edges of the blindspot was 3.7 degrees and 8.6 degrees respectively. In all participants the blindspot covered the location 15˚ into the temporal hemifield and 1˚ inferior of fixation.

## Procedure

Participants completed at least three sessions on different days. Participants viewed the display monocularly with the preferred eye. The non-preferred eye was occluded using custom made glasses. In the first session participants received 3 blocks of 80 training trials during which no eye-movements were made and the presentation duration of the discrimination target was reduced from 250ms (block 1), to 150ms (block 2) and finally 60ms (block 3). They then performed two blocks of 80 trials in the Fixate condition. Participants were instructed to ignore the saccade target and maintain gaze at the fixation point. Trials began with the onset of the fixation point. After a variable latency (750-1500ms) the saccade target appeared. The discrimination target was presented 100 or 150ms after the onset of the saccade target. After 60ms checkerboards appeared to mask the discrimination target and the location contralateral to the discrimination target. These masks remained present until a response was made. The discrimination target appeared with equal probability at each of the four locations. Participants then completed the Saccade condition in which they were instructed to make a vertical saccade to fixate of the saccade target. Otherwise the procedure was the same as in the Fixate condition. Participants received two blocks of 96 training trials in which the presentation duration of the discrimination target was reduced from 150ms (Block 1) to 60ms (Block 2). They then

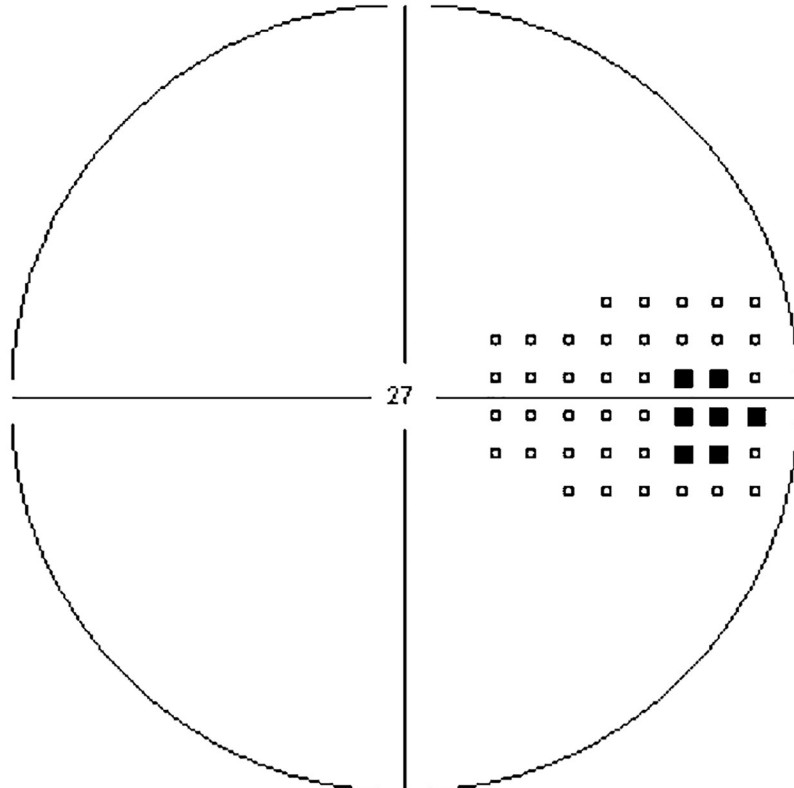

**Fig 1. Sample visual field plot for a right-eyed participant.** The mapped area extended from 5˚ to 19˚ into the temporal hemispace and 5˚ superior to 5˚ inferior of the horizontal midline. Open squares show normal vision, closed black squares show areas of blindness. Each square represents 1.72˚ visual angle.

completed at least 8 blocks of 96 experimental trials, spread out over several experimental sessions. The different trial types were presented in a random order within each block. Fig 2 illustrates the sequence of events in a typical Saccade trial. A twelve-point calibration was performed prior to each block of trials.

## Selection criteria for eye-movements

Eye-movements were selected offline for 6 participants and online for 2 participants. Online saccades were selected when the eye moved beyond the boundary of a 2 x 2 degree RoI around fixation and landed in 4 x 4 degree RoI centred on the saccade target. Offline saccade selection used a velocity criterion of 70˚/s for a minimum duration of 20ms. Tracking failure resulted in the loss of 2.5% trials. The remaining trials were filtered to remove those where a saccade was made to the probe location (7.8% trials), the saccade was hypometric by more than 2˚ (4% of trials) and trials where the saccade was initiated before the discrimination target was masked (12.2% trials).

## Results

### Target discrimination

Only trials in which a saccade was executed between 0 and 300 ms after the onset of the mask were included in the analysis. A repeated measures ANOVA with factors of Discrimination

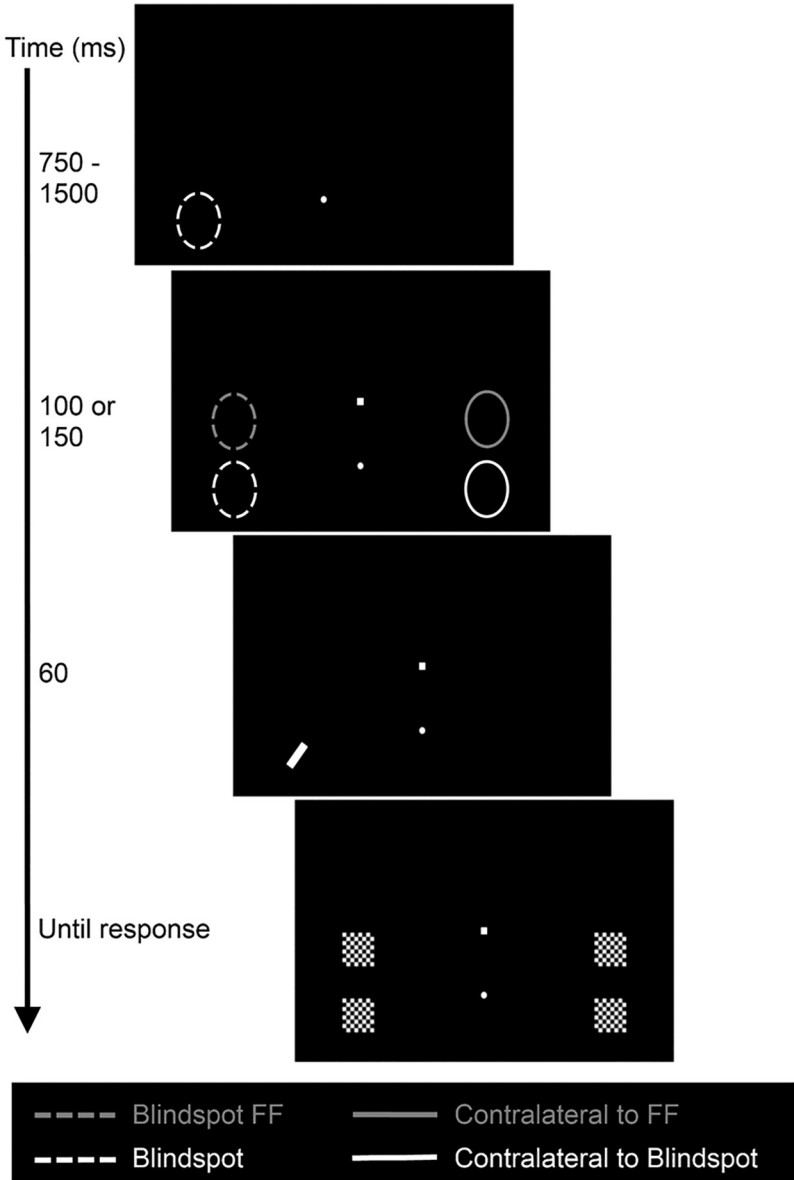

**Fig 2. Schematic illustration of a typical trial.** The ovals illustrate the potential locations of the discrimination target and were not visible to the participant. Times are in milliseconds.

Target Location (Blindspot, Blindspot Future Field, Contralateral to Blindspot, Contralateral to Blindspot Future Field) and Condition (Fixate, Saccade) revealed an interaction between Discrimination Target Location and Condition ($F_{(3,21)}$ = 6.9, $p$ = .002, partial eta$^2$ = .5 see Fig 3). Holm-Bonferroni corrected paired samples t-tests comparing performance at each stimulus location at each level of condition showed that performance at the Blindspot FF was significantly worse in the Saccade condition than in the Fixate condition (64% vs 82%, $t_{(7)}$ = 4.33, $p$ < .01, $d$ = 1.53). None of the 3 other comparisons reached statistical significance. A further post-hoc t-test demonstrated discrimination performance at the Blindspot FF was significantly impaired relative to Blindspot Contra FF in the saccade condition (64% vs 84%, $t_{(7)}$ = 4.08, $p$ <

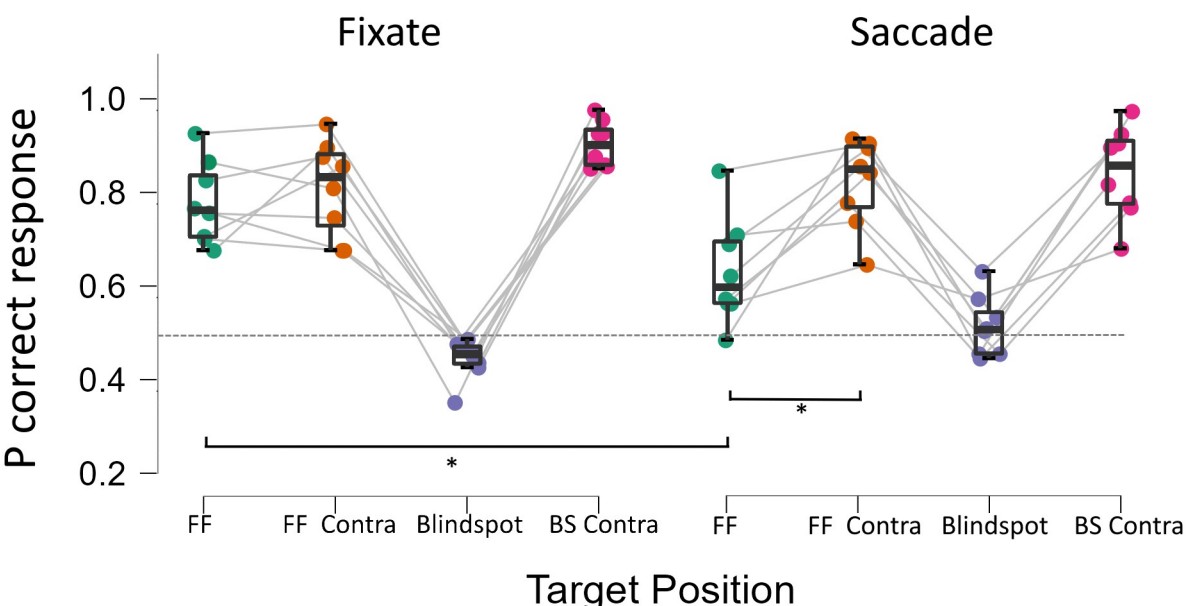

**Fig 3. Discrimination accuracy in saccade and fixate conditions.** Boxes show mean probability of correct identification at each stimulus position in the Fixate and Saccade conditions. Each disc shows data from each participant, the dashed horizontal line shows chance level of performance. Asterisks show p < .01. Error bars show 95% *CI*.

.01, *d* = 1.44). In contrast, in the Fixate condition there was no significant difference between Blindspot FF and Blindspot Contra FF (80% vs83%, $t_{(7)}$ = .41, *p* = .41, *d* = .31).

Presaccadic effects typically become more pronounced with proximity to saccade onset. The data were therefore split into two 140ms bins; a proximal bin of trials in which the probe was presented between 60 and 200ms before saccade onset and a distal bin in which the probe appeared between 201 and 340ms before saccade onset. There were an average of 69 trials per location in the proximal bin (inter-individual range 25–122) and 55 trials per location in the distal bin (inter-individual range 20–125). Note the inclusion criteria were extended to include trials executed up to 340ms after mask onset to ensure there were at least 20 trials in each bin in the 'distal' condition. A planned comparison (paired samples t-test) revealed that performance was significantly worse in the proximal bin (61.3%) than the distal bin (67.8%; $t_{(7)}$ = 3.08, *p* = .018, *d* = 1.1), indicating that the presaccadic impairment was greater closer to saccade onset (see Fig 4).

## Saccade metrics

The mean saccade amplitude was 5.4˚ (range 5.1˚-5.8˚), the mean saccade latency was 279 ms (range 196–407) and the mean peak velocity was 287 m/s (range 198–363). The mean amplitude, peak velocity and latency of eye-movements were analysed using one-way repeated measures ANOVAs with a factor of Stimulus Position (FF, FF Contra, Blindspot and Blindspot Contra). There was no effect of Stimulus Position on Amplitude (*F* = 0.68, *p* = .577; partial eta$^2$ = .12), Latency (*F* = 0.45, *p* = .72, partial eta$^2$ = .08) or Peak Velocity (*F* = 0.69, *p* = .72, partial eta$^2$ = .09).

## Discussion

This experiment examined how the intention to make a saccade affected pre-saccadic perception of stimuli that were briefly presented at the post-saccadic location of the blindspot (the

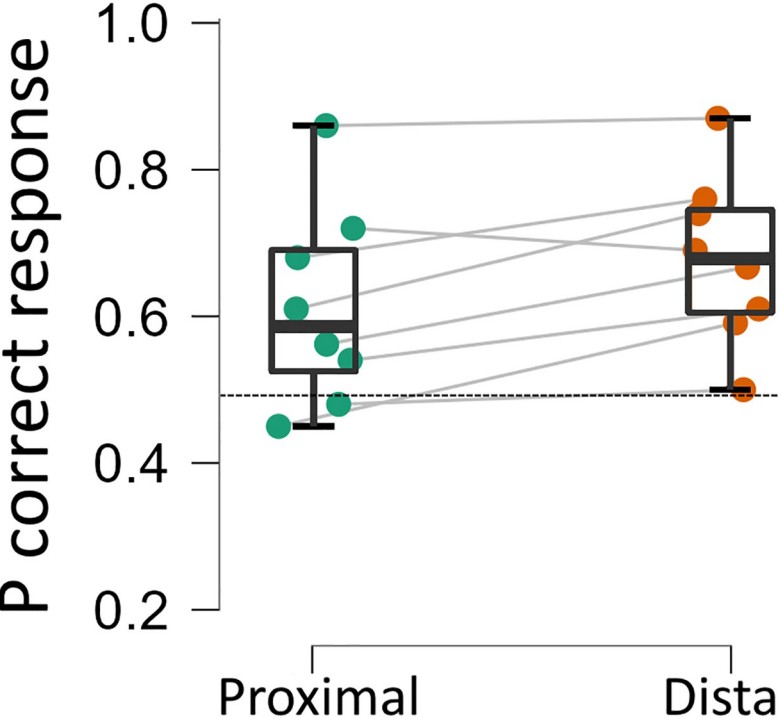

**Fig 4. Discrimination accuracy in proximal and distal conditions.** Boxes show mean probability of correct response at the Blindspot Future Field target position in the proximal (discrimination target offset 0-140ms prior to saccade onset) and distal (target offset 141- 280ms before saccade onset) time bins during the Saccade condition. Each disc shows data from a single participant, the dashed horizontal line shows chance level of performance. Error bars show 95% CI.

Bindspot Future Field, FF). Discrimination accuracy at Blindspot FF was significantly lower in the saccade condition relative to the fixation condition and this impairment was greater when the probe appeared closer in time to saccade onset. These data suggest that there is spatially specific decrement in perceptual processing at the blindspot FF that is dependent on the intention to make a saccade.

The presaccadic perceptual impairment at the blindspot FF may be related to the presaccadic 'forward remapping' of receptive fields [9, 15, 23]. In these studies, some cells appear to have their receptive fields remapped before a saccade, such that they begin responding to stimuli that will appear in their post-saccadic receptive field. This remapping can begin up to 400ms before saccade onset [10], and peaks in the 100-50ms prior to saccade onset (e.g. [1]), a time-course that shows similarities with the effects observed here. One might object that the blindspot, by definition, has no visual receptive fields so cannot be remapped. However, [22] have shown although there are no direct retinal inputs from the blindspot to primary visual cortex, there are V1 neurons with receptive fields that represent the blindspot. Furthermore, the phenomenon of perceptual filling in (the illusory perception of a feature such as colour or luminance in the blindspot when the feature occurs around the periphery of the blindspot) is associated with activation of these cells [24]. If this 'visuotopic' organisation is preserved at higher levels of the visual system then presaccadic remapping could lead to the region of space containing the target being represented by cells with receptive fields with no visual inputs, thus disrupting the ability to perceive the orientation of the probe.

The finding that the blindspot is remapped across a saccade may have implications for understanding and treatment of hemianopia, particularly in cases where there is post-chiasmatic but pre-cortical damage to the optic nerve. In this case, the patient will have preserved retinotopic maps in V1 that no longer receive input from the eye, similar to the representation of the blindspot. Hemianopes typically make hypometric and disorganised eye movements [25] which are hard to explain purely in terms of their sensory deficit. However, if the blind field of a hemianope is remapped in a way analogous to the blind spot of a healthy participant it may be adaptive to make many short saccades into the intact field, as it minimises the volume of space in which visual sensitivity is reduced. Given that people typically make ~3 saccades every second, one implication of our data is that, under normal viewing conditions, hemianopes may have functional impairment that extends from their blind field into the endpoint of a saccade in the good field, and which would go undetected by standard measures of visual function in which fixation is enforced. This ipsilateral functional impairment during eye-movements may help explain why hemianopia is so resistant to rehabilitation, and why transfer from successful lab-based interventions to real-world improvements in visual ability has been so hard to achieve [21, 26, 27].

To summarize, there is a significant pre-saccadic perceptual impairment for stimuli presented at the post-saccadic location of the blindspot. This perceptual impairment follows a similar time-course to the presaccadic remapping of visual receptive fields and may have implications for understanding the problems with rehabilitation experienced by people with acquired scotomas.

## Author Contributions

**Conceptualization:** Daniel T. Smith.

**Data curation:** Daniel T. Smith.

**Formal analysis:** Daniel T. Smith, Mark Avery.

**Investigation:** Mark Avery.

**Methodology:** Daniel T. Smith.

**Resources:** Daniel T. Smith.

**Software:** Daniel T. Smith.

**Supervision:** Daniel T. Smith.

**Visualization:** Daniel T. Smith.

**Writing – original draft:** Daniel T. Smith, Ulrik Beierholm, Mark Avery.

**Writing – review & editing:** Daniel T. Smith, Ulrik Beierholm, Mark Avery.

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
