## [Decision Letter · Decision Letter 0]

3 Aug 2023

PONE-D-23-17473A presaccadic perceptual impairment at the postsaccadic location of the blindspotPLOS ONE

Dear Dr. Smith,

Thank you for submitting your manuscript to PLOS ONE. After careful consideration, we feel that it has merit but does not fully meet PLOS ONE’s publication criteria as it currently stands. Therefore, we invite you to submit a revised version of the manuscript that addresses the points raised during the review process.

We look forward to receiving your revised manuscript.

Kind regards,

Shenbing Kuang, Ph.D.

Academic Editor

PLOS ONE

Journal Requirements:

3. Please ensure that you refer to Figure 1 in your text as, if accepted, production will need this reference to link the reader to the figure.

5. We note that Supporting Figures [screenshot of email & screenshot of document datestamp] in your submission contain copyrighted images. All PLOS content is published under the Creative Commons Attribution License (CC BY 4.0), which means that the manuscript, images, and Supporting Information files will be freely available online, and any third party is permitted to access, download, copy, distribute, and use these materials in any way, even commercially, with proper attribution. For more information, see our copyright guidelines: http://journals.plos.org/plosone/s/licenses-and-copyright.

(1) You may seek permission from the original copyright holder of Supporting Figures [screenshot of email & screenshot of document datestamp] to publish the content specifically under the CC BY 4.0 license. 

(2) If you are unable to obtain permission from the original copyright holder to publish these figures under the CC BY 4.0 license or if the copyright holder’s requirements are incompatible with the CC BY 4.0 license, please either i) remove the figure or ii) supply a replacement figure that complies with the CC BY 4.0 license. Please check copyright information on all replacement figures and update the figure caption with source information. 

If applicable, please specify in the figure caption text when a figure is similar but not identical to the original image and is therefore for illustrative purposes only.

**Additional Editor Comments:**

Your manuscript has been reviewed by two experts in the field. Both reviewers are generally very positive about your study and the manuscript. Meanwhile, they have also provided a few minor comments and suggestions to further improve the clarity of your manuscript.

Reviewers' comments:

Reviewer's Responses to Questions

**Comments to the Author**

1. Is the manuscript technically sound, and do the data support the conclusions?

Reviewer #1: Yes

Reviewer #2: Yes

2. Has the statistical analysis been performed appropriately and rigorously? 

Reviewer #1: No

Reviewer #2: Yes

3. Have the authors made all data underlying the findings in their manuscript fully available?

Reviewer #1: No

Reviewer #2: Yes

4. Is the manuscript presented in an intelligible fashion and written in standard English?

Reviewer #1: Yes

Reviewer #2: Yes

5. Review Comments to the Author

Reviewer #1: Review of Research article PONE-D-23-17473 « A presaccadic perceptual impairment at the postsaccadic location of the blindspot »

Participants viewed stimuli monocularly and were required to make vertical eye-movements. A discrimination target is presented before saccade execution at 4 possible locations : within the blindspot, contralateral to the blindspot, in post-saccadic location of the blindspot and contralateral to the post-saccadic location of the blindspot. A significant perceptual impairment is revealed not only at the blindspot location, but also at the post-saccadic location of the blindspot. A control condition with eye fixed is provided with only a deficit at the blindspot. This is a puzzling result which is very important to confirm that there is a pre-saccadic remapping process that can be distinguished from regular (covert) attention. These two processes have been shown to be dissociated in a single-case patient (Pre-saccadic perceptual facilitation can occur without covert orienting of attention. Blangero A, Khan AZ, Salemme R, Deubel H, Schneider WX, Rode G, Vighetto A, Rossetti Y, Pisella L. Cortex. 2010 Oct;46(9):1132-7. doi: 10.1016/j.cortex.2009.06.014. Epub 2009 Jul 15. PMID: 19660745 )

I only have minor comments :

* Line 161 : « before the saccade goal was masked », the saccade goal is never masked, do you mean the discrimination target instead ?

*It is written in the abstract that « There was a significant perceptual impairment at the post-saccadic location of the blindspot relative to the contralateral post-saccadic location of the blindspot and the post-saccadic location of the blindspot in a no-saccade control condition ». It seems then contradictory to read line 169 : « Holm-Bonferroni corrected paired samples t-tests showed that performance at the Blindspot FF was significantly worse in the Saccade condition than in the Fixate condition (64.% vs 82%, t(7) = 4.33, p < .01, d = 1.53). No other comparison reached statistical significance ». May be the stats mentioned in the abstract correspond to ANOVAs and post-hoc tests made with separated temporal bins ? Please provide these post-hoc tests, together with a graph illustrating this secondary analysis.

*Line 175 : As I understand, the ranges mentioned here are inter-individual range of the number of trials in each temporal bin, may be it should be mentioned.

*Line 215 : « The finding that that », please correct the typo

*Line 215 : « many have implications », please correct the typo, many should be « may »

*Line 216 : spell out what AMD initials mean, age macular degeneration ? This is a deficit concerning central vision, so I do not clearly see how the result of this paper could be related to it ? Speculation about hemianopia is more relevant and would be interesting to test. Furthermore, hemianopia could come from different lesions, at the level of V1 or at the level of the optic nerve after the optic chiasma. It could be interesting to contrast these two levels.

Reviewer #2: This is a paper about a psychophysical study of visual perception around the time of saccades. The specific question is whether perisaccadic remapping, whereby receptive fields change position in advance of a saccade, includes the retinal blind spot. This is important for determining whether remapping comes with a perceptual downside and whether this downside might account for problems observed in people with vision loss.

It's a clever experiment, well-designed and presented, and the results seem clear enough. I have a few suggestions for revision, mostly cosmetic:

1) It would generally help to have a bit more information in the figure captions, though I was mostly able to figure out what was shown. The exception was figure 1. What is shown in the left panel? How large a portion of the visual field is represented by each square in the right panel? The text doesn’t refer to the figure at all.

2) What was the contrast of the discrimination target? How was it chosen?

3) My main suggestion is to include more information about how the timing of events in the experiment was verified. This is obviously critical, because the reported results can arise trivially if the display of the probe persists into the postsaccadic period. This can happen in two ways:

a. Filtering/smoothing of the eye position traces often lead to inaccurate estimates of saccade latency. Please indicate how this was done.

b. Persistence of the stimulus is a well-known problem, particular with bright stimuli on a dark background in CRT displays (see the back and forth on this topic in the early 1980s, e.g., Irwin, Yantis, and Jonides (1983)). I doubt this is a problem for the current work, because the authors wisely used a mask to eliminate any potential persistence. But it seems like there were some trials in which the offset of the probe occurred very close to the saccade, and it would be good to make sure that small timing errors did not influence the results.

4) Related to the previous point, I didn’t entirely understand the timing. What was the actual distribution of probe timing relative to saccade onset? The bins described in the paragraph around line 171 seem arbitrary and not entirely consistent with the reported range of saccade latencies and the experiment timing shown in Figure 2.

5) The authors might be interested in a few older papers on visual receptive fields covering the blind spot from Komatsu’s group – see for example Matsumoto and Komatsu (2005).

6. PLOS authors have the option to publish the peer review history of their article (what does this mean?). If published, this will include your full peer review and any attached files.

Reviewer #1: **Yes: **PISELLA Laure

Reviewer #2: **Yes: **Christopher Pack

---

## [Author Response · Author response to Decision Letter 0]

23 Aug 2023

EDITOR

We would like to thank you and the reviewers for their thoughtful and constructive feedback on our manuscript. We have revised the manuscript to try and address all the points raised in their reviews. The response to reviewers is detailed in separate document, and our response to the journal requirements in outlined below:

1. Please ensure that your manuscript meets PLOS ONE's style requirements

Completed 

2. In your Data Availability statement, you have not specified where the minimal data set underlying the results described in your manuscript can be found. PLOS defines a study's minimal data set as the underlying data used to reach the conclusions drawn in the manuscript and any additional data required to replicate the reported study findings in their entirety. Upon re-submitting your revised manuscript, please upload your study’s minimal underlying data set as either Supporting Information files or to a stable, public repository and include the relevant URLs, DOIs, or accession numbers within your revised cover letter. For a list of acceptable repositories, please see http://journals.plos.org/plosone/s/data-availability#loc-recommended-repositories. Any potentially identifying patient information must be fully anonymized.

The underlying dataset can be found of OSF https://osf.io/c47te/

3. Please ensure that you refer to Figure 1 in your text as, if accepted, production will need this reference to link the reader to the figure.

Line 113 now has a reference to Fig 1

We have removed the supporting information as it is not germane to the research reported in the article 

5. We note that Supporting Figures [screenshot of email & screenshot of document datestamp] in your submission contain copyrighted images. All PLOS content is published under the Creative Commons Attribution License (CC BY 4.0), which means that the manuscript, images, and Supporting Information files will be freely available online, and any third party is permitted to access, download, copy, distribute, and use these materials in any way, even commercially, with proper attribution. For more information, see our copyright guidelines: http://journals.plos.org/plosone/s/licenses-and-copyright.We require you to either (1) present written permission from the copyright holder to publish these figures specifically under the CC BY 4.0 license, or (2) remove the figures from your submission:

These files have been removed

Citations have been reviewed as requested.

REVIEWERS

* Line 161 : « before the saccade goal was masked », the saccade goal is never masked, do you mean the discrimination target instead ?

Thank you for spotting this, we did indeed mean the discrimination target. This has been corrected.

*It is written in the abstract that « There was a significant perceptual impairment at the post-saccadic location of the blindspot relative to the contralateral post-saccadic location of the blindspot and the post-saccadic location of the blindspot in a no-saccade control condition ». It seems then contradictory to read line 169 : « Holm-Bonferroni corrected paired samples t-tests showed that performance at the Blindspot FF was significantly worse in the Saccade condition than in the Fixate condition (64.% vs 82%, t(7) = 4.33, p < .01, d = 1.53). No other comparison reached statistical significance ». May be the stats mentioned in the abstract correspond to ANOVAs and post-hoc tests made with separated temporal bins ? Please provide these post-hoc tests…

Thanks for pointing this out. When reporting the results on (line 169), we intended to refer to the outcome of 4 paired-samples t-tests comparing performance at each of the 4 locations in the saccade and fixate conditions, not the full suite of post-hoc tests. We agree this is potentially misleading as the reader might reasonably assume that ‘no other tests…’ implies that we ran all possible combinations of posthoc tests. We had also intended to report the contrast between the blindspot FF and the Contralateral to Blindspot Future Field separately, but accidentally omitted it. We have therefore edited the text (lines 164-177) as to be more precise in the reporting of the results, and include the analysis comparing Blindspot FF with Contralateral to Blindspot Future Field referred to in the abstract. We have also revised figure 3 to indicate the significant posthocs. 

…together with a graph illustrating this secondary analysis.

Figure 4 has been added to illustrate the posthoc test

*Line 175 : As I understand, the ranges mentioned here are inter-individual range of the number of trials in each temporal bin, may be it should be mentioned.

Line 180 edited as suggested

*Line 215 : « The finding that that », please correct the typo

Corrected

*Line 215 : « many have implications », please correct the typo, many should be « may »

Corrected 

*Line 216 : spell out what AMD initials mean, age macular degeneration ? This is a deficit concerning central vision, so I do not clearly see how the result of this paper could be related to it ? Speculation about hemianopia is more relevant and would be interesting to test. Furthermore, hemianopia could come from different lesions, at the level of V1 or at the level of the optic nerve after the optic chiasma. It could be interesting to contrast these two levels.

"The original idea of including discussion of AMD (age macular degeneration) was to ask what happens to presaccadic attention when the fovea is blind. On reflection we agree this is probably not that relevant to the current experiment, as the issue with AMD is retinal not cortical. We have therefore removed reference to AMD. 

The suggestion to compare hemianopes with V1 vs damage to the optic chiasm is really interesting. It seems to us that the patients with post-chiasmatic optic nerve damage have a deficit that is most similar to the blindspot, as there remains a retinotopically organised region in V1 that does not receive any visual input. It is however less certain what the predictions are for patients with cortical damage. In this case the cortical retinotopy will have broken down, and it is not clear how this will affect the remapping process. We have added a couple of lines to the discussion to make this point.

Reviewer #2: It's a clever experiment, well-designed and presented, and the results seem clear enough. I have a few suggestions for revision, mostly cosmetic:

Thank you for the positive feedback

1) It would generally help to have a bit more information in the figure captions, though I was mostly able to figure out what was shown. The exception was figure 1. What is shown in the left panel? How large a portion of the visual field is represented by each square in the right panel? The text doesn’t refer to the figure at all.

The left and right panel show the same data, but the right panel illustrates performance at each of the tested locations, whereas the left panel shows a probability plot. As the information in the left panel is redundant this panel has been removed. Details on the size of the square have been added to the caption, and a reference to the figure added to line 119

2) What was the contrast of the discrimination target? How was it chosen?

The target was white on a black background with approximately 100% contrast. We selected a high contrast target as we needed something that would be discriminable even at quite large distances from fovea, but would also fall fully within the blindspot. The best way to achieve this was to use relatively small but high contrast target. 

3) My main suggestion is to include more information about how the timing of events in the experiment was verified. This is obviously critical, because the reported results can arise trivially if the display of the probe persists into the postsaccadic period. This can happen in two ways:

a. Filtering/smoothing of the eye position traces often lead to inaccurate estimates of saccade latency. Please indicate how this was done.

We agree timing is critical. The selection algorithm smoothed over 4 samples and used a criteria of a minimum velocity of 70 d/s and duration of 20ms to detect eye-movements. 

b. Persistence of the stimulus is a well-known problem, particular with bright stimuli on a dark background in CRT displays (see the back and forth on this topic in the early 1980s, e.g., Irwin, Yantis, and Jonides (1983)). I doubt this is a problem for the current work, because the authors wisely used a mask to eliminate any potential persistence. But it seems like there were some trials in which the offset of the probe occurred very close to the saccade, and it would be good to make sure that small timing errors did not influence the results.

We excluded any trials where the saccade began before the onset of the mask. Given that a 5 degree saccade has a duration of 20-30ms, we feel this is rather conservative selection criterion that guarantees that the blindspot cannot physically have overlapped the target position on any trial. Nevertheless, we also looked at the 123 trials in which the saccade began between 1 and 10ms after the onset of the mask, of which 23 were trials where the target appeared in the Blindspot FF. Mean accuracy in the blindspot FF was 60% on these trials. Given the very small number of trials we feel confident that, even if the occasional trial with small timing errors slipped through the filtering, the main finding cannot be explained away by these trials. It is also worth noting that our second analysis indicates that perceptual impairment is present in the distal bin, which cannot possibly be affected by small errors in timing that would mean the blindspot was occluding the target position before the mask appeared. 

4) Related to the previous point, I didn’t entirely understand the timing. What was the actual distribution of probe timing relative to saccade onset? The bins described in the paragraph around line 171 seem arbitrary and not entirely consistent with the reported range of saccade latencies and the experiment timing shown in Figure 2.

We thank the reviewer for pointing this out. We had omitted a line explaining the inclusion criteria for trials in the first analysis, which we agree makes the discussion of the timing a bit hard to understand. To clarify: For the first analysis we excluded all trials where the saccade began before the mask was presented. We included trials where a saccade was executed between 0 and 300 ms of the onset of the mask. Note this is slightly different to the second analysis, in which we included trials in which saccades were executed up to 340ms after saccade onset. The reviewer is correct that the bins were somewhat arbitrary as we were aiming for bins of the same temporal size and a minimum of 20 trials per bin. We have revised the results section (lines 165-187) to try and make this more explicit to the reader. 

5) The authors might be interested in a few older papers on visual receptive fields covering the blind spot from Komatsu’s group – see for example Matsumoto and Komatsu (2005).

We thank the reviewer for this suggestion and have added a line about this study to the discussion

---

## [Decision Letter · Decision Letter 1]

1 Sep 2023

A presaccadic perceptual impairment at the postsaccadic location of the blindspot

PONE-D-23-17473R1

Dear Dr. Smith,

We’re pleased to inform you that your manuscript has been judged scientifically suitable for publication and will be formally accepted for publication once it meets all outstanding technical requirements.

Kind regards,

Shenbing Kuang, Ph.D.

Academic Editor

PLOS ONE

Additional Editor Comments (optional):

Reviewers' comments:

Reviewer's Responses to Questions

**Comments to the Author**

1. If the authors have adequately addressed your comments raised in a previous round of review and you feel that this manuscript is now acceptable for publication, you may indicate that here to bypass the “Comments to the Author” section, enter your conflict of interest statement in the “Confidential to Editor” section, and submit your "Accept" recommendation.

Reviewer #1: All comments have been addressed

Reviewer #2: All comments have been addressed

2. Is the manuscript technically sound, and do the data support the conclusions?

Reviewer #1: Yes

Reviewer #2: (No Response)

3. Has the statistical analysis been performed appropriately and rigorously? 

Reviewer #1: Yes

Reviewer #2: (No Response)

4. Have the authors made all data underlying the findings in their manuscript fully available?

Reviewer #1: Yes

Reviewer #2: (No Response)

5. Is the manuscript presented in an intelligible fashion and written in standard English?

Reviewer #1: Yes

Reviewer #2: (No Response)

6. Review Comments to the Author

Reviewer #1: I am satisfied by this revised version

Thank you

Reviewer #2: (No Response)

7. PLOS authors have the option to publish the peer review history of their article (what does this mean?). If published, this will include your full peer review and any attached files.

Reviewer #1: No

Reviewer #2: **Yes: **Christopher Pack

---

## [Editor Report · Acceptance letter]

5 Sep 2023

PONE-D-23-17473R1 

A presaccadic perceptual impairment at the postsaccadic location of the blindspot 

Dear Dr. Smith:

I'm pleased to inform you that your manuscript has been deemed suitable for publication in PLOS ONE. Congratulations! Your manuscript is now with our production department. 

Kind regards, 

on behalf of

Associate Professor Shenbing Kuang 

Academic Editor

PLOS ONE